# Quantitative Analysis of Indoor Gaseous Semi-Volatile Organic Compounds Using Solid-Phase Microextraction: Active Sampling and Calibration

**Jianping Cao [1,2]** , **Li Zhang [1]** , **Zhibin Cheng [1]** , **Siqi Xie [1]** , **Runze Li [1]** , **Ying Xu [3]** and **Haibao Huang [1,2,*]**

[1] School of Environmental Science and Engineering, Sun Yat-sen University, Guangzhou 510006, China; caojp3@mail.sysu.edu.cn (J.C.); zhangli66@mail2.sysu.edu.cn (L.Z.); chengzhb3@mail2.sysu.edu.cn (Z.C.); xiesq5@mail2.sysu.edu.cn (S.X.); lirunze3@mail2.sysu.edu.cn (R.L.)

[2] Guangdong Provincial Key Laboratory of Environmental Pollution Control and Remediation Technology, Guangzhou 510006, China

[3] Department of Building Science, Tsinghua University, Beijing 100084, China; xu-ying@mail.tsinghua.edu.cn

[*] Correspondence: huanghb6@mail.sysu.edu.cn; Tel.: +86-20-3933-6475

**Abstract:** Semi-volatile organic compounds (SVOCs) are important pollutants in indoor environments. Quantification of gaseous SVOC concentrations is essential to assess the pollution levels. Solid-phase microextraction (SPME) is considered to be an attractive sampling technique with merits, including simplicity of use, rapid sampling, and solvent free. However, the applications of SPME for sampling gaseous SVOCs are often limited by the fluctuating velocity of indoor air (leading to an unstable sampling rate) and the uncertainties associated with the traditional calibration of SPME. Therefore, we established an SPME-based active sampler to ensure the stable sampling of SVOCs in fluctuating air and developed a two-step calibration method based on the sampling principle of SPME. The presented method and a traditional method (sorbent tubes packed with Tenax TA) were simultaneously used to measure SVOC concentrations in an airstream generated in experiments. Three typical indoor SVOCs, diisobutyl phthalate (DiBP), tris (1-chloro-2-propyl) phosphate (TCPP), and benzyl butyl phthalate (BBzP) were chosen as the analytes. Mean concentrations measured by SPME agreed well with the sorbent tubes (relative deviations <12%), supporting the feasibility of the presented method. Further studies are expected to facilitate the application of the presented method (especially the problem associated with the sampling-tube loss of low volatile SVOCs).

**Keywords:** indoor air quality; semi-volatile organic compounds; solid-phase microextraction; active sampling; gas-phase concentration; chemical analysis

## 1. Introduction

Semi-volatile organic compounds (SVOCs) are ubiquitous in indoor environments [1–4]. Human exposure to certain SVOCs has been found to be associated with adverse health effects, including asthma [5], endocrine disruption [6], reproductive problems [7], and even cancer [8]. Accurate quantification of SVOC concentrations in the indoor air is essential to assess exposure levels and the associated health risks [2].

SVOCs often exhibit low concentrations in the indoor air, typically <5 μg/m³ (air volume), due to their low vapor pressures and strong partitioning from the gas phase to surfaces [1,2]. Enrichment is therefore indispensable for quantifying gaseous SVOC concentrations. During the enrichment, gaseous SVOCs are collected in sampling media, such as Tenax TA, polyurethane foam (PUF), adsorption resin (XAD-resin), polyethylene, and their combinations [9]. Except for Tenax TA, solvent extraction is often used to desorb SVOCs from the sampling media to organic solvents [9]. Solvent extraction has several disadvantages, e.g., laborious, time-consuming (e.g., several hours), and solvent-consuming. Tenax TA is solvent free, so the collected SVOCs can be thermally desorbed and then directly

injected into the gas chromatograph (GC). However, high temperatures (e.g., 300 °C) and relatively long desorption times (e.g., 30 min) are often required to ensure high desorption efficiency [10]. Additionally, the cost of the instrument used for thermal desorption is expensive. Since the early 1990s, the use of solid-phase microextraction (SPME) for sampling SVOCs in the air has been reported in some studies [11–13]. SPME has several merits: solvent free, simplicity of use (direct injection to GC), and relatively short analysis time (e.g., 5 min thermal desorption in the GC injection port) [14,15]. These merits become significant for sampling gaseous SVOCs.

SPME mainly comprises a plunger and a sampling fiber attached to the plunger, as described in Section S1 of the Supporting Information (SI). The sampling fiber is a cylindrical fused silica fiber with a thin polymeric coating surrounding it. When sampling, SPME is exposed to the sample matrix, and the analytes are sorbed by the coating [14]. Typically, the sorption process is required to reach equilibrium because it facilitates the calibration of SPME. After equilibrium has been reached, the analyte concentration ($C$) can be easily determined based on the amount of the analytes sorbed by the SPME coating ($M$), the volume of the SPME coating ($V$), and a pre-determined equilibrium constant ($K$), i.e., $C = M/(K \cdot V)$ [15]. However, the time required to reach equilibrium sorption can be very long for gaseous SVOCs, e.g., 141 h for di(2-ethylhexyl) phthalate (DEHP) [13], which is infeasible in applications. Furthermore, competitive adsorption may occur if multiple analytes coexist in the sample matrix [16]. Therefore, non-equilibrium sampling methods were established, and kinetic models were used to calibrate SPME (relating $M$ to $C$ and the sampling time) [15,17]. The mass-transfer rate of analytes from the sample matrix to the SPME coating (designated as $R_m$) is a key parameter in the kinetic models, which can be significantly affected by the velocity of the sample matrix flowing across the SPME coating [11,12,15–20]. Careful control of the flow velocity across the coating is required to maintain a constant $R_m$ because the ambient air velocity is always fluctuating and uncontrollable [15]. In existing applications, SPME was typically used as a passive sampler by directly placing SPME in the sample matrix [21]. However, the effect of the fluctuating air flow on $R_m$ cannot be eliminated in the passive sampling mode. In some existing studies, SPME has been used to sample gaseous volatile organic compounds (VOCs) in the active sampling mode [21,22]. Active sampling allows an air flow to pass over the SPME fiber at a stable rate, which is helpful in obtaining a stable sampling rate. Currently, the SPME-based quantitative analysis of gaseous SVOCs have only been successfully implemented in either static air (zero air velocity) [16,17,23,24] or streaming air where the air velocity is well controlled and monitored (in chamber studies) [11,13,19,20]. SPME was used in passive sampling mode in these studies.

Most applications of SPME for quantitative analysis require knowing the absolute value of $M$. In existing studies, the instrument response of the analytes sampled by SPME is often converted to the absolute value of $M$ based on the calibration curve obtained by the injections of liquid standards to GC [15]. The transfer efficiency of analytes from the GC injector to the GC column is assumed to be similar for SPME and liquid injections [15]. However, several studies have proven that the transfer efficiencies could be significantly different between SPME and liquid injections, leading to large uncertainties in the calibration of SPME based on liquid standards [25]. As an alternative, direct-loading methods were developed to calibrate SPME by manually adding known amounts of liquid standards to the SPME fiber [26,27]. Nevertheless, the careful and skillful operation is required to accurately control the loaded amount and avoid damaging the breakable SPME fiber. More accurate calibration methods of SPME are required.

The objectives of this study are, therefore, to (i) design a device to enable the stable SPME sampling of indoor SVOCs in the fluctuating air, (ii) develop an accurate calibration method of SPME, and (iii) evaluate the performance of the newly-developed device and calibration method.

## 2. Materials and Methods

### 2.1. SPME-Based Active Sampler

In this study, we designed an SPME-based active sampler (designated as SPME-AS) mainly composed of a two-ferrule tee connector, as illustrated in Figure 1.

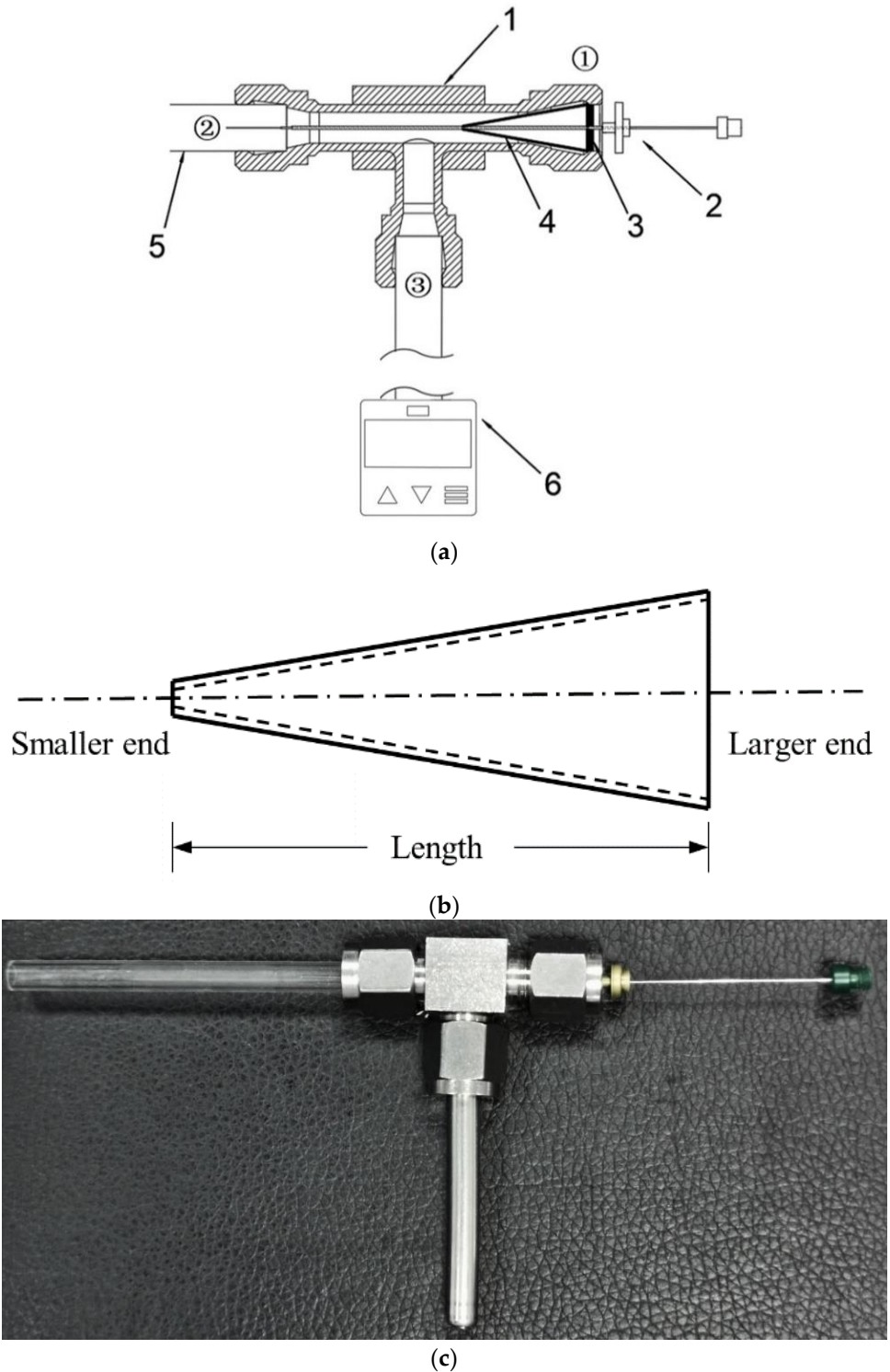

**Figure 1.** (**a**) Schematic of the SPME-based active sampler designed in this study. (**b**) Schematic of the truncated cone tube in the SPME-based active sampler and (**c**) photo. 1, two-ferrule tee connector; 2, SPME; 3, sealing gasket; 4, truncated cone; 5, sampling tube; and 6, sampling pump. ①, ②, and ③ correspond to openings 1, 2, and 3, respectively.

As shown, one of the axial openings of the tee connector (opening 1) was blocked by a sealing gasket, the other axial opening (opening 2) was connected to a short tube (designated as sampling tube), and the vertical opening (opening 3) was connected to a sampling pump. The tee connector had two ferrules at each opening, front ferrule and back ferrule. The front ferrule of opening 1 was replaced by the sealing gasket. In order to ensure that the sampling fiber of SPME was located exactly on the central axis of the sampling tube, the back ferrule of opening 1 was replaced by a truncated cone tube (see details in Figure 1b). The diameter of the smaller end of the cone tube was slightly greater than the diameter of the stainless-steel microtube of SPME, and the diameter of the larger end equaled that of the back ferrule. The cone angle of the cone tube was equal to that of the back ferrule. When sampling, the sampling pump kept running at a constant rate, and the air flowing through the SPME coating could be controlled at a constant velocity.

In this study, the tee connector was an equal tee used for connecting tubes with an external diameter of 6 mm, which was made of stainless steel (complied with GB/T 3745-2008 of China: 24° Cone Connectors–Union Tee (in Chinese)). An Agilent inlet septum (Agilent Tech., Part No. 5183-4757) was used as the sealing gasket. The sampling tube was made of silica glass with an external diameter of 6 mm, internal diameter of 4 mm, and length of 6 cm. SPME was purchased from Sigma-Aldrich Co. LLC. (Supelco Analytical, Cat. No. 57302). The fiber coating was made of polydimethylsiloxane, with a length of 1 cm and thickness of 7 μm. Details about the SPME used in this study can be found in SI Section S1. The truncated cone tube was made of polypropylene (free of SVOCs) with a length of 12.5 mm, diameter of 1.1 mm at the smaller end, diameter of 6 mm at the larger end, and thickness of 0.5 mm for the tube wall. The inlet of the sampling tube had a distance of 3.2 cm to the very end of the sampling fiber of SPME.

*2.2. Calibration Method of SPME*

Typically, the calibration of SPME includes two steps: (1) transform the response of the analytical instrument (e.g., peak area, designated as *A*) to the amount of the analytes sorbed on SPME coating (*M*) and (2) transform *M* to the analyte concentration in the sample matrix. In some scenarios, the above two steps can be combined by preparing standard sample matrices to directly link the instrument responses to the analyte concentrations [15]. However, the one-step calibration curves often vary over time since the instrument status is changing. According to our measurements, the one-step calibration curve may exhibit a change of over 50% within one week. Therefore, the one-step calibration requires frequent updating, which may be inconvenient in applications because the SPME sampling of standard matrices needs to be repeated several times (e.g., 5 times) in every calibration process, and each SPME sampling may take minutes to hours. A new two-step calibration method was therefore established in this study.

**Step 1. Transform the instrument responses to the sampled amount**. The liquid injection, although its transfer efficiency differs from SPME, is still employed because it can be used to reflect the changes in the instrument status and can be automatically run. For liquid injections, the instrument response ($A_l$) is always proportional to the injected amount of the analyte ($M_l$),

$$M_l = k_l \cdot A_l \tag{1}$$

where $k_l$ is the slope of the liquid-based calibration line. $k_l$ can be obtained by measuring $A_l$ of a series of liquid standards (with different $M_l$) and then fitting Equation (1) to the measured points.

For SPME injections, a similar relationship exists between the instrument response ($A_s$) and the sampled amount of SPME ($M_s$),

$$M_s = k_s \cdot A_s \tag{2}$$

where $k_s$ is a constant. In practical applications, $k_s$ is always unknown because $M_s$ is unknown. Alternatively, the SPME samples are treated as the liquid samples, and Equation (1) is USED to obtain an "equivalent" sampled amount ($M_s^l$) after $A_s$ is obtained,

$$M_s^l = k_l \cdot A_s \tag{3}$$

Substituting Equation (2) into Equation (3), we get a linear relationship between the "equivalent" sampled amount and the real sampled amount of SPME,

$$M_s^l = k_l \frac{M_s}{k_s} = k \cdot M_S \tag{4}$$

where $k$ equals $k_l/k_s$. $k_l$ and $k_s$ are both affected by the transfer efficiency of the analytes and the status of the analytical instrument. According to the principle of the internal standard method (widely used in quantitative analysis), the change of the instrument status should have equal effects on $k_l$ and $k_s$. Therefore, $k$ actually represents the ratio of the transfer efficiency of the liquid samples to that of the SPME samples. $k$ is supposed to be a constant if the assembly of the analytical instrument is unchanged (e.g., the liner of the injector) [25].

**Step 2. Transform the "equivalent" sampled amount to the concentration**. As introduced in SI Section S1 (Figure S2), the sorption process of the analytes in the coating of SPME can be divided into three regimes: (1) linear regime (the amount of the analytes sorbed by SPME linearly increases as the sampling time increases), (2) kinetic regime (the increasing rate of the sorption amount gradually decreases as the sampling time increases), and (3) equilibrium regime (the sorption amount reaches equilibrium) [15]. Typically, the SPME-based sampling process is limited to either the equilibrium regime or the linear regime [15]. However, as mentioned above, the equilibrium regime is not suitable for gaseous SVOCs because it may require a long sampling time. Therefore, the linear regime is preferable for the sampling of gaseous SVOCs by SPME.

In the linear regime, $M_s$ is proportional to the sampling time ($t$) and the gaseous SVOC concentration ($C_g$) [20,23],

$$M_S = h_m \cdot S \cdot C_g \cdot t \tag{5}$$

where $h_m$ (m/s) is the convective mass-transfer coefficient of SVOCs over the coating surface, and $S$ (m$^2$) is the surface area of the coating, which is a known constant for a given SPME. $h_m$ is treated as an unknown constant because it is difficult to be accurately measured or estimated.

Substituting Equation (4) into Equation (5), we can get:

$$M_s^l = k \cdot h_m \cdot S \cdot C_g \cdot t = \beta \cdot C_g \cdot t \tag{6}$$

where $\beta$ is the product of $k$, $h_m$, and $S$. That says, the two unknown constants ($k$ and $h_m$) are merged into one unknown constant ($\beta$, the calibration constant of SPME). $\beta$ has a unit of m$^3$/s.

In summary, $k_l$ and $\beta$ are the two unknown constants that need to be determined for calibrating SPME. First, the instrument responses of a series of liquid standards are measured, and then $k_l$ is determined by fitting Equation (1) to the measured points. Second, the instrument responses of a series of SPME samples (sampling gaseous SVOCs with known and constant concentration over a series of times in the linear regime) are measured, and the corresponding $M_s^l$ is determined based on Equation (3), and then $\beta$ is determined by fitting Equation (6) to the data points (several pairs of sampling time and $M_s^l$).

In the new calibration method, the problem corresponding to the different transport efficiencies between SPME and liquid injections has been considered in the calibration constant ($\beta$). In addition, the instrument status should have no effects on $\beta$ since both $k$ and $h_m$ are not related to the instrument status. Therefore, only one single measurement is needed to determine $\beta$ if the assemblies of SPME-AS and the analytical instrument are kept unchanged. This merit greatly simplifies the calibration of SPME. Certainly, periodic

update of $k_l$ (e.g., per week) is still necessary to correct the changes in instrument status, which is, however, convenient to obtain.

When analyzing the SPME samples, the instrument response is transformed to the "equivalent" sampled amount by Equation (3) and the newest $k_l$, and then the "equivalent" sampled amount is transformed to the gaseous SVOC concentration by Equation (6), $\beta$, and the sampling time. Therefore, the exact amount of SVOCs sampled by SPME is no longer a must in both the calibration and the quantitative analysis.

### 2.3. Experimental System

In order to evaluate the feasibility of the newly-developed SPME-AS and calibration method, we designed an experimental system, as illustrated in Figure 2. A stainless-steel chamber (in the shape of a circular tube) was used to generate an SVOC-laden airstream with stable SVOC concentrations. The chamber was coated inside by carefully pulling a piece of pure-cotton gauze soaked with pure SVOC liquid through it to form a thin SVOC layer on the inner wall. As fresh air (free of SVOCs) is introduced into the chamber, the air flow will carry gaseous SVOCs (which emit from the SVOC layer to the air flow) toward the chamber outlet. The gaseous SVOC concentration at the chamber outlet was found to equal the saturated gaseous concentrations of the corresponding SVOC for a source chamber with an inner diameter of 17.2 mm, length of 30.0 cm, and air flow rate lower than 100 mL/min [28,29]. Therefore, the inner diameter and length of the present source chamber were chosen to be 17.2 mm and 50.0 cm, respectively, and the air flow rate was controlled to be 75 mL/min. It was confirmed that the film of SVOC liquid on the chamber wall was stable during the experiments (the variation of gaseous SVOC concentration at the chamber outlet before and after our experiments was found to be less than 12%).

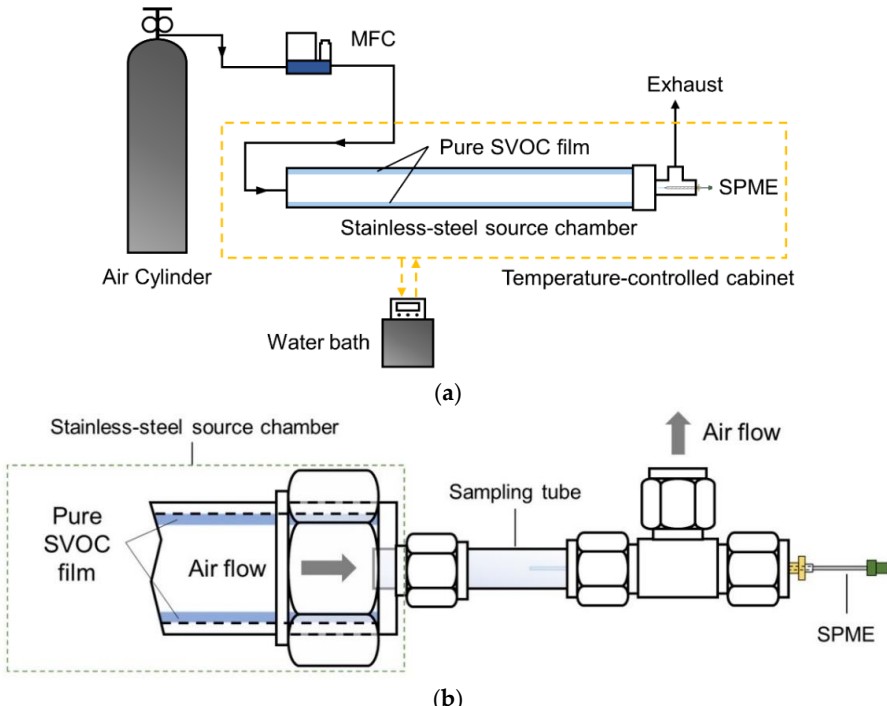

**Figure 2.** Illustration of (**a**) the experimental system and (**b**) the connection of the SPME-based active sampler to the source chamber.

The SPME-AS was attached to the outlet of the source chamber, as illustrated in Figure 2b. In this way, the SVOC-laden air will flow into the sampling tube, and then gaseous SVOCs will be sorbed by the SPME coating. The front end of the sampling tube should be aligned with the outlet of the source chamber to avoid the sorption of gaseous SVOCs by the tube connector [30]. In addition, a traditional sampler, a sorbent tube packed

with Tenax TA, was employed to quantify the SVOC concentrations at the outlet of the source chamber [30]. The sorbent tube was also directly attached to the outlet of the source chamber. According to our previous study, the flow rate through the sorbent tube was suggested to be lower than 100 mL/min to ensure high capture efficiency (>95%) of SVOCs [31]. This condition was also satisfied by choosing a flow rate of 75 mL/min in the experiments. The sorbent tubes were purchased from Markes International Ltd. (Part number C1-AAXX-5003).

A gas cylinder with high purity nitrogen (>99.9%, no SVOC and water vapor) was used as the supply air. Mass flow controller (MFC, MC-1SLPM, Alicat Scientific, Tucson, AZ, USA) upstream of the source chamber was used to control the flow rate of the airstream through the source chamber, SPME-AS, and the sorbent tube. The source chamber and the samplers were placed in a temperature-controlled cabinet (the temperature was controlled using a water bath with a precision of 0.5 °C). Note, the sampling pump of SPME-AS was not necessary for the present experimental system because the sampling flow can be well controlled by the MFC.

*2.4. Experimental Procedure*

Two experiments were conducted to quantify the calibration constant $\beta$ and evaluate the performance of the new sampler and calibration method. Two SVOCs commonly found in indoor environments, diisobutyl phthalate (DiBP) and tris(1-chloro-2-propyl) phosphate (TCPP), were the target analytes in the first-stage experiments. DiBP and TCPP are widely used as plasticizer and flame retardant, respectively, in indoor materials and products [32–35]. The experiments of DiBP and TCPP were separately conducted.

**Experiment 1**. Measuring the calibration constant $\beta$ by the following steps:

Step 1: The gaseous SVOC concentrations at the outlet of the source chamber were measured using sorbent tubes. The measurements were repeated four times. The sampling time was set to 5–30 min. Detailed sampling times of the sorbent tubes can be found in SI Section S2 (Table S1).

Step 2: An SPME-AS (without SPME) was connected to the outlet of the source chamber, and the SVOC-laden air flow was allowed through the sampler for one hour. Note, strong sorption of SVOCs by the inner surface of the sampling tube of SPME-AS was found in our pre-experiments. For DiBP and TCPP, one hour was proven to be enough to eliminate the effects of the sampling-tube loss. Details about the effect of sampling-tube loss on the quantitative analysis of gaseous SVOCs can be found in our another study of us [36].

Step 3: One SPME was inserted into the SPME-AS through the sealing gasket, pushing out the sampling fiber from the SPME plunger. After a certain sampling time, the SPME was removed from the sampler, and the "equivalent" sorption amount of SVOC was determined in the coating of SPME. The sampling time of SPME was set to 15–600 s. Details about the sampling times of SPME can also be found in SI Section S2 (Table S2).

Step 4: Step 3 was repeated 5 times. Note, different SPMEs and sampling times were used in each repetition.

Step 5: $\beta \cdot C_g$ was determined by fitting Equation (6) to the measured data, then calculating the calibration constant $\beta$ using the $C_g$ measured in Step 1.

Step 6: The experimental temperature was changed, and repeating steps 1–5. The experiments were conducted at three temperatures (20 °C, 25 °C, and 30 °C).

**Experiment 2**. Evaluating the performance by the following steps:

Step 1: Steps 1 and 2 of "experiment 1" were repeated.

Step 2: One SPME was inserted into the sampler, and the target SVOC was sampled for a certain time. The sampling times of DiBP and TCPP were set to 120 s and 300 s, respectively.

Step 3: The "equivalent" sorption amount of SVOC in the coating of SPME was determined and then transformed to the gaseous SVOC concentration by combining Equation (6), the calibration constant $\beta$ (determined in "experiment 1"), and the sampling time.

Step 4: Steps 2 and 3 were repeated 3 times.

Step 5: The results of SPME were compared to those of sorbent tubes to evaluate the accuracy and stability of the SPME method.

Step 6: The experimental temperature was changed, and repeating steps 1–4. The experiments were conducted at two temperatures (23 °C and 27 °C).

All surfaces of the source chamber and the sampler were rinsed with dichloromethane (DCM) prior to the experiments. Origin 2018 (OriginLab Corporation, Northampton, MA, USA) was employed for curve (line) fitting in this study.

### 2.5. Chemical Analysis

After sampling, the surface of the stainless-steel rod of SPME, which would also adsorb SVOCs, was carefully wiped three times using pure-cotton gauze soaked with DCM. The SPME was then analyzed using a gas chromatography-mass spectrometry system (GC-MS, Agilent Technologies 8890 GC system equipped with a 5977B Mass Selective Detector) by manually inserting SPME into the front injector of the GC (280 °C), and thermally desorbing the SPME coating for 5 min. Prior to the experiments, each SPME was conditioned by thermal desorption in the GC injector for 5 min to ensure that no SVOCs remained in the SPME coating. No SVOCs could be detected after one-time conditioning. Note: the 5 min desorption of SPME may not be long enough for other SVOCs with higher molecular weight. The sorbent tube was analyzed using a thermal desorber (UNITY-xr, Markes International) connected to the back injector of the above GC-MS. The sorbent tubes were conditioned at 325 °C for 40 min with high purity nitrogen at 100 mL/min flow rate and sealed tightly prior to use. After conditioning, the amount of SVOCs that remained in the sorbent tubes was treated as negligible (<10 ng) because they were over 20 times lower than the amount of SVOCs collected in the sorbent after sampling (>200 ng).

Details of the analysis protocols of SPME and sorbent tubes, the calibration of GC-MS for sorbent tubes, the calibration of GC-MS by injecting liquid standards (i.e., determining $k_l$), and chemicals used in the experiments can be found in SI Section S3.

## 3. Results and Discussion

### 3.1. Gaseous Concentrations Measured by Sorbent Tubes

The key to our experiments is to know the gaseous concentrations ($C_g$) of DiBP and TCPP in the generated airstream. $C_g$'s measured by sorbent tubes packed with Tenax TA are listed in Table 1. The measurements of $C_g$ were repeated four times at each temperature. The relative standard deviation ($RSD$) of $C_g$ was lower than 20% for each case, which was consistent with that reported in the existing study (using a similar source chamber and sorbent tubes, $RSD$ ranged from 0.1% to 20%) [28]. The relatively low $RSD$ indicated that the concentrations of DiBP and TCPP at the outlet of the source chamber were stable. This is an important precondition for the following results.

**Table 1.** Comparison of the gaseous concentrations ($C_g$) of DiBP and TCPP measured by the sorbent tubes (packed with Tenax TA) with those measured by the SPME-based active sampler.

| | DiBP | | | | | TCPP | | | | |
|---|---|---|---|---|---|---|---|---|---|---|
| | Tenax TA | | SPME | | | Tenax TA | | SPME | | |
| $T$ (°C) | $C_g$ (μg/m³) [a] | $RSD$ [b] | $C_g$ (μg/m³) [a] | $RSD$ [b] | $RD$ [c] | $C_g$ (μg/m³) [a] | $RSD$ [b] | $C_g$ (μg/m³) [a] | $RSD$ [b] | $RD$ [c] |
| 20 | 246 | 9.5% | – [d] | – | – | 250 | 19% | – | – | – |
| 23 | 372 | 20% | 353 | 5.0% | 5.1% | 341 | 8.6% | 355 | 11% | 4.1% |
| 25 | 456 | 8.0% | – | – | – | 520 | 3.7% | – | – | – |
| 27 | 586 | 19% | 575 | 2.3% | 1.9% | 566 | 16% | 664 | 2.3% | 17% |
| 30 | 767 | 9.8% | – | – | – | 1035 | 11% | – | – | – |

[a] $C_g$ is the average gaseous concentration of four time measurements. [b] $RSD$ is the relative standard deviation of $C_g$ obtained by four time measurements. [c] $RD$ is the relative deviation between $C_g$ measured by Tenax TA sorbent tubes and $C_g$ measured by SPME ($|C_{g\_Tenax} - C_{g\_SPME}|/C_{g\_Tenax} \times 100\%$). [d] $C_g$ not measured by SPME method because corresponding experiments were only conducted at 23 °C and 27 °C.

In our experiments, high purity SVOC liquids were used. Thus, $C_g$ at the outlet of the source chamber should be equal to the saturated gaseous concentrations ($C_{sat}$) of the corresponding pure SVOC liquids, i.e., $C_g = C_{sat}$. However, the availability of reliable values of $C_{sat}$ is limited for many SVOCs because significant differences (by several orders of magnitude) can be found in the literature [30]. Taking DiBP as an example, the reported values of $C_{sat}$ at 25 °C were in the range of 57 µg/m$^3$ to 8649 µg/m$^3$, as summarized by Wu et al. [28] (corresponding to 510 Pa to 77,000 Pa in Wu et al.). $C_{sat}$ of DiBP measured in this study was around 50% lower than that measured by Wu et al. [28] (456 µg/m$^3$ vs. 1076 µg/m$^3$ at 25 °C). Although the discrepancy is somewhat significant, they are at least on the same order of magnitude.

As indicated in Table 1, $C_g$ increases significantly with increasing temperature. A well-known equation, the Clausius−Clapeyron equation, is often used to describe the temperature dependence of $C_{sat}$ [37]. As discussed in SI Section S4, the measured results agreed well with the Clausius−Clapeyron equation. The consistency supports, to some extent, the reliability of $C_g$ measured by the sorbent tubes.

### 3.2. Calibration Constant of SPME (β)

The results of "experiment 1" are shown in Figure 3. It should be noted that the measurements were all conducted according to the time series determined in SI Section S2 (Table S2), i.e., 15–300 s for DiBP and 30–600 s for TCPP. However, the sampled amount of SVOCs in SPME coating might exceed the upper limits of the calibration line of GC-MS in some cases (30 ng for DiBP and 40 ng for TCPP, see details in SI Section S3). Those data points were not shown in Figure 3 because the results obtained by GC-MS might not be accurate above the upper limits. Equation (6) was used to fit the remaining points.

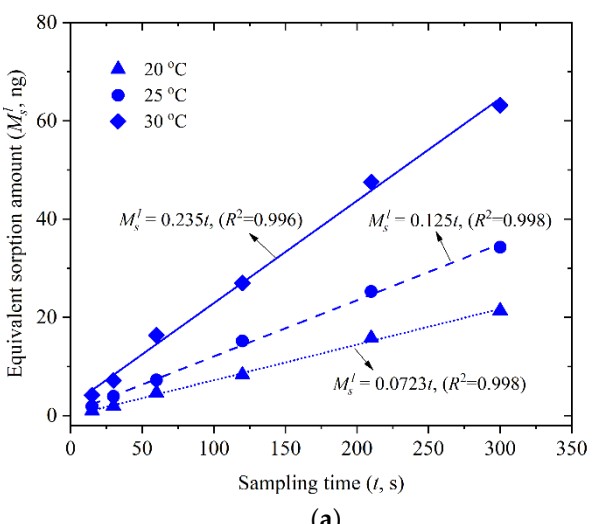
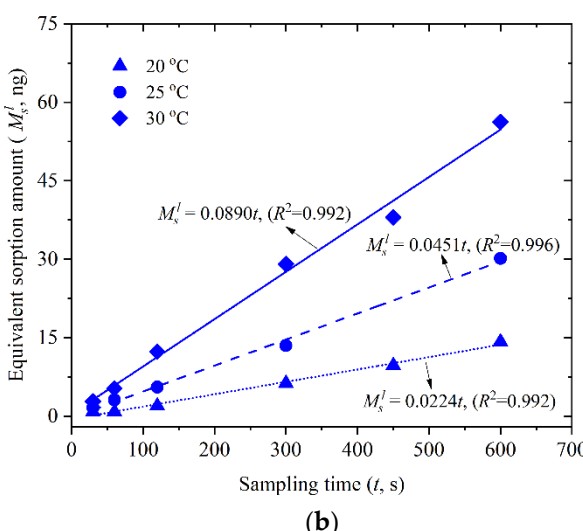

(**a**)  (**b**)

**Figure 3.** Comparison between experiment results and the lines fitted using Equation (6) for (**a**) DiBP and (**b**) TCPP. $R^2$ is the square of the correlation coefficient of linear fitting. Slope of the fitted line divided by the corresponding gaseous concentration equals the SPME calibration constant (β).

As shown in Figure 3, the experimental data agreed with the fitted lines in all cases ($R^2 > 0.99$), validating the use of Equation (6). In addition, linear curve fitting without forcing the intercept to zero was also conducted. In all cases, the obtained intercepts were close to zero, i.e., at least 10 times lower than the minimum sampling amount in the corresponding case. Furthermore, the relative deviations between the slope of the theoretical equation (Equation (6)) and the slope of the fitted line without forcing the intercept to zeros were all less than 10%. The insignificant intercepts and small relative deviations of the slopes further support the use of Equation (6).

The slope of the fitted line was used to estimate the calibration constant of SPME ($\beta$) by dividing the slope by the corresponding $C_g$ measured by sorbent tubes (listed in Table 1). Table 2 lists the values of $\beta$ for DiBP and TCPP. It can be seen that the values of $\beta$ were quite stable even though the experiments were conducted at different temperatures (20–30 °C). The *RSD* of $\beta$ was lower than 6% for both SVOCs. In addition, the relative deviations between $\beta$ of a certain temperature and the average $\beta$ of three temperatures were less than 7% and 3% for DiBP and TCPP, respectively. Theoretically, $\beta$ tends to increase with increasing temperature because $h_m$ is a function of the diffusivity of SVOCs in the air ($D_a$) [38], and $D_a$ is positively related to the air temperature [37]. Typically, $h_m$ is proportional to $D_a^{4/3}$ and $D_a$ is proportional to $T^{1.75}$ ($T$ is the temperature, K) [37,38]. Thus, $h_m$ (or $\beta$, $\beta = k \times h_m$ and $k$ is independent to $T$) is proportional to $T^{7/3}$ ($4/3 \times 1.75$). Therefore, $\beta$ will increase by 8% if the temperature increases from 20 °C (293 K) to 30 °C (303 K), the change of $\beta$ is likely offset by the experimental errors (<7%, see above). Additionally, our measurements indicated that the $\beta$'s of DiBP and TCPP were relatively stable over a three-month period (see details in SI Section S5), supporting the assumption that the instrument status has no effect on $\beta$.

**Table 2.** SPME calibration constants ($\beta$) of DiBP and TCPP measured at different temperatures.

| SVOCs | Temperature (°C) | $\beta_T \times 10^4$ (m³/s) [a] | $\beta \times 10^4$ (m³/s) [b] | *RSD* (%) [c] |
|---|---|---|---|---|
| DiBP | 20 | 2.94 | 2.92 | 5.6 |
| | 25 | 2.74 | | |
| | 30 | 3.07 | | |
| TCPP | 20 | 0.895 | 0.874 | 2.1 |
| | 25 | 0.867 | | |
| | 30 | 0.860 | | |

[a] $\beta_T$ is the calibration constant of SPME measured at a certain temperature. [b] $\beta$ is the average calibration constant of SPME measured at three temperatures. [c] *RSD* is the relative standard deviation of $\beta$ measured at three temperatures.

In summary, the temperature variation (20–30 °C) may insignificantly affect the calibration constant of SPME. $\beta$ determined at a certain temperature can be directly used for the quantitative analysis of the SPME samples, provided that the indoor temperature varies in a relatively narrow range (e.g., 25 ± 5 °C). Certainly, it is still necessary to experimentally verify the stability of $\beta$ at a wider temperature (e.g., 0–30 °C).

### 3.3. Comparison between SPME and Sorbent Tubes

Table 1 also lists the results of "experiment 2", i.e., the comparison of $C_g$ measured by the sorbent tubes and $C_g$ measured by SPME-AS at 23 °C and 27 °C. In "experiment 2", the "equivalent" sampled amount of SPME was converted to the value of $C_g$ using the average $\beta$ determined above. The $C_g$ values measured by SPME-AS were very similar to those measured by the sorbent tubes, with relative deviations less than 6% and 5% for DiBP and TCPP, respectively. In addition, we compared the sampling precision (stability) between the sorbent tubes and SPME-AS. The *RSD*s of $C_g$ obtained by SPME-AS (four time measurements) were less than 5% and 11% for DiBP and TCPP, respectively. Meanwhile, the *RSD*s of the sorbent tubes (four time measurements) were in the ranges of 8–20% and 3.7–19% for DiBP and TCPP, respectively. The *RSD*s of SPME-AS tended to be lower than those of the sorbent tubes (especially for DiBP). The highly consistent between SPME-AS and sorbent tubes and lower *RSD* of SPME-AS support the feasibility of SPME-AS for the quantitative analysis of gaseous SVOCs.

### 3.4. Application of SPME to Low Volatile SVOCs

The strong sorption of SVOCs by the inner surface of the sampling tube of SPME-AS was noticed in the experiments. Thus, $C_g$ in the air flow reaching the sampling fiber of SPME may be lower than $C_g$ in the sample air. In the experiments of DiBP and TCPP, the

sample air was controlled to flow through the sampling tube for one hour to eliminate the effects of the sampling tube loss. However, the one-hour sorption may not be enough for SVOCs with lower volatility. According to another study of us [36], the time required to reach equilibrium sorption might be longer than a day for some SVOCs, e.g., 27 h for benzyl butyl phthalate (BBzP). Therefore, the above sampling procedure is not applicable for low volatile SVOCs.

An alternative experiment was conducted to preliminarily evaluate the applicability of SPME-AS for low volatile SVOCs. BBzP was chosen as the target SVOC (another typical plasticizer in indoor environments [1]). Details about the experiments and results are provided in SI Section S6. Similar to DiBP and TCPP, the results indicated that temperature had insignificant effects on $\beta$ (*RSD* of $\beta$ measured at three temperatures was lower than 15%, and the relative deviation between $\beta$ of a certain temperature and the average $\beta$ of three temperatures was less than 17%). $C_g$ of BBzP measured by SPME-AS was consistent with the sorbent tubes (the relative deviation between them was lower than 12%), indicating acceptable accuracy of SPME-AS for BBzP. However, the lower stability of $C_g$ measured by SPME was observed. The *RSD* of $C_g$ measured by SPME-AS was in the range of 15–30%, while the *RSD* of sorbent tubes was still similar to those of DiBP and TCPP (7–18%). The reduction in the sampling stability of SPME may be because the sampling tube of SPME-AS was replaced in each measurement, and nuances in the roughness of the inner surfaces may exist among sampling tubes (leading to different sorption ability of BBzP).

## 4. Discussions

There are several limitations in the present study. First, the sorption of SVOCs by the inner surface of the sampling tube strongly limits the stability of SPME-AS. This limit can be reduced, to some extent, by involving the effects of the sampling-tube loss of gaseous SVOCs in the calibration constant of SPME (as employed in "experiment 3" and discussed in SI Section S6) or replacing the silica-glass sampling tube with tubes made by materials with low sorption capacity of SVOCs, e.g., deactivated glass or Teflon. Nevertheless, the tube-sorption problem may still exist for SVOCs with extremely low volatility. Detailed discussion about the sampling-tube loss can be found in another study of us [36].

Second, the performance evaluation (accuracy and stability) of SPME-AS was only accomplished by laboratory measurements. The generated SVOC-laden air was quite different from the realistic indoor air. The key differences included high purity nitrogen, no airborne particles, no humidity, constant air flow, single species of SVOC, and high gaseous SVOC concentrations. It is still unclear whether similar performance can be obtained for the sampling in realistic indoor environments. In addition, the present method has been applied to only three species of SVOCs; its general applicability to other SVOCs is completely unclear. Furthermore, the competitive sorption on the fiber coating of SPME and the sampling tube may occur if multiple SVOCs coexist [16]; the effects of competitive sorption on the sampling time of SPME and the sampling tube loss require further study.

Third, the present design of SPME-AS may be far from optimal. With the goal of highest accuracy and stability, the flow rate through the sampling tube, the inner diameter of the sampling tube, the length of the sampling tube (or the distance from the inlet of the sampling tube to SPME), the sampling time of SPME, and the material of sampling tube can be optimized in the future.

## 5. Conclusions

In this study, an SPME-based active sampler (SPME-AS) was developed to facilitate the sampling of SVOCs in the fluctuating air in indoor environments. A two-step calibration method was established for the accurate calibration of SPME based on the sampling principle of SPME-AS. The SPME method (SPME-AS combined with the calibration method) was applied to measure the concentrations of three typical SVOCs (DiBP, TCPP, and BBzP) in an air stream generated by a specially-designed chamber. The measured concentrations were found to be highly consistent with the measurements of a traditional method widely

used for sampling gaseous SVOCs in indoor environments (sorbent tube packed with Tenax TA). Relative deviations between the SPME method and the Tenax TA method were less than 12% for all cases. In addition, the temperature was found to have an insignificant effect on the calibration constant of SPME, which facilitates the application of SPME-AS in indoor environments with unstable temperatures. For SVOCs with higher volatility (DiBP and TCPP), lower RSDs were observed with the SPME method as compared to the Tenax TA method, supporting the stability of the SPME method. The results preliminarily demonstrated that the proposed SPME method is feasible for sampling and analyzing SVOCs in the indoor air. However, strong sorption of gaseous SVOCs by the inner surface of the sampling tube of SPME-AS was noticed in the experiments, which may induce significant errors in the SPME method. The effects of sampling tube loss are especially important for SVOCs with lower volatility. Further studies are required to characterize and reduce the effects of the sampling tube loss for SPME-AS.

**Supplementary Materials:** The following supporting information can be downloaded at: https://www.mdpi.com/article/10.3390/atmos13050693/s1, Section S1: Details of SPME; Section S2: Sampling times of sorbent tubes and SPME; Section S3: Chemical analysis; Section S4: Temperature dependence of the measured gaseous concentrations; Section S5: Time variation of the SPME calibration constant; Section S6: Experiments for low volatile SVOC (BBzP). References [13,15,17,28,37,39–41] are cited in the supplementary materials.

**Author Contributions:** Conceptualization, J.C., Y.X. and H.H; methodology, J.C., L.Z., Z.C., S.X. and H.H.; investigation, J.C., L.Z., Z.C., S.X., R.L., Y.X. and H.H.; experiments, J.C., L.Z., Z.C., S.X. and R.L.; data analysis, J.C., Z.C., S.X. and R.L.; writing—original draft preparation, J.C.; writing—review and editing, L.Z., Z.C., S.X., R.L., Y.X. and H.H. All authors have read and agreed to the published version of the manuscript.

**Funding:** This research was funded by the Natural Science Foundation of China (grant number 51908563), Guangdong Basic and Applied Basic Research Foundation (grant number 2019A1515011179), Science and Technology Program of Guangzhou (grant number 202102020990), and Beijing Key Laboratory of Indoor Air Quality Evaluation and Control (grant number BZ0344KF20-11).

**Institutional Review Board Statement:** Not applicable.

**Informed Consent Statement:** Not applicable.

**Data Availability Statement:** Not applicable.

**Conflicts of Interest:** The authors declare no conflict of interest.

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
