# Peer review of "Quantitative Analysis of Indoor Gaseous Semi-Volatile Organic Compounds Using Solid-Phase Microextraction: Active Sampling and Calibration"

_atmosphere, doi:10.3390/atmos13050693_

Round 1

Reviewer 1 Report

In this study, the authors developed an SPME-based active sampler to ensure the stable sampling of SVOCs in the fluctuating air in indoor environments. This novel device was well designed and had a reliable calibration method. The performance of this device has been verified by the traditional Tenax TA method. This manuscript should be warranted for publication. Some suggestions are as follows:
1. Line 223, how to ensure that the coated SVOC film was stable during the experiments?
2. Whether the authors have compared inserting the sampling fiber directly into the source chamber, so as to avoid the error caused by the adsorption of SVOC on the inner surface of the sampling tube.
3. In this study, the experimental system has a constant flow rate and the environment is relatively ideal. The characteristics of this device cannot be fully displayed. Therefore, its reliability can be further tested in a dynamic environment in the future.

Author Response

Please find the responses in the attached file.

Reviewer 2 Report

In the present study (atmosphere-1693136), entitled “Quantitative analysis of indoor gaseous semi-volatile organic compounds using solid-phase microextraction: Active sampling and calibration” a SPME-based active sampling method was developed to analyze SVOCs in fluctuating air. The study covers a two-step calibration method that was used for the determination of diisobutyl phthalate (DiBP), tris(1-chloro-2-propyl) phosphate (TCPP) and benzyl butyl phthalate (BBzP) in the indoor environment. These compounds were determined by a custom-made active sampling cell and its sampling theory was also studied. The manuscript is generally well written and can be accepted after major revision. Some points are listed below.

  • Please revise the following Sentences:

Line 3:” upstream the sampler” should be corrected as “upstream of the sampler”

Line 57: That says, the analyte concentration (C) can be easily determined by the amount of the analytes sorbed by the SPME coating (M) and a pre-determined equilibrium constant conditions

Line 243: This is another reason why we chosen the flow rate of 75 mL/min in the experiments.

Line 296, Chemical Analysis: Cleaning of the surface of stainless steel cover with DCN cannot be suitable. Therefore, the effect cleaning process should be controlled.

Line 199, Supplementary file: “For DiBP, the cold trap was purged with a helium flow rate of 31.5 mL/min, 1.5 mL/min of the flow was transferred to the GC column, the other 30 mL/min was exhausted. For TCPP, the cold trap was purged with a helium flow rate of 51.2 mL/min, 1.2 mL/min of the flow was transferred to the GC column, the other 50 mL/min was exhausted.”

  • In the supplementary files, different sections and figure numbers have been used. The order in which the supplementary files are presented in the manuscript is not the same as in the supplementary files. This makes it difficult to follow the manuscript. Please remove the section numbers from the supplementary files and rearrange the number of the supplementary file according to the order of the manuscript.
  • The water bath presentation in Figure 2 is not clear. Please control the figure.
  • SI Table S1: It should be explained why different sampling times are used at different temperatures.
  • Line 263, Step 2: The authors noted that sVOC adsorption of the sampling tube did not interfere after 1 hour of exposure. This also extends the analysis time. However, this conditioning time can be varying according to the concentration of the analyte. This also extends the analysis time. Why deactivation of glass or the use of the usage of inert material wasn't preferred for the sampling tube?
  • Line 210, 2.3. Experimental System: Experimental System: A stainless-steel chamber was used for producing SVOC gas standards. The constant sVOC emission of the system is very important for accurate measurement. Therefore, the change of analyte concentration against time should be monitored for a longer period and the experiments should be carried out at the constant region.
  • The sampling time for each analyte was studied in the absence of each other. But the accuracy of the results can be changed due to the competitive adsorption of analytes. Therefore, for more accurate results, sampling times should be optimized in the presence of all analytes.
  • The concentration unit in Table 1 should be presented.
  • In the thermal desorption experiment, it should be checked whether all the analytes passing through the tube are adsorbed or not.
  • Line 337: Table 2 does not match the sentence.
  • There are 3 variables in Figure 3 but two markings were used for each analyte. Please use only color or shape single marking for each analyte.
  • Line 204, Suplementary File: The following sentence is not clear “For liquid standards, 1 μL of the solution was injected into the front injector of GC (liner: 5190-3163, Agilent Tech).”
  • Calibration curves of the analytes should be performed in the concentration units such as mg/m3.

Author Response

(The authors gave the same response as above.)

Round 2

Reviewer 2 Report

 In the present study (atmosphere-1693136), entitled “Quantitative analysis of indoor gaseous semi-volatile organic compounds using solid-phase microextraction: Active sampling and calibration” required corrections have been made in the manuscript and it can be accepted in its current form.